# GSK-3α Inhibition in Drug-Resistant CML Cells Promotes Susceptibility to NK Cell-Mediated Lysis in an NKG2D- and NKp30-Dependent Manner

**DOI:** 10.3390/cancers13081802

**Published:** 2021-04-09

**Authors:** Nayoung Kim, Mi Yeon Kim, Woo Seon Choi, Eunbi Yi, Hyo Jung Lee, Hun Sik Kim

**Affiliations:** 1Department of Convergence Medicine, University of Ulsan College of Medicine, Seoul 05505, Korea; naykim@amc.seoul.kr; 2Asan Institute for Life Sciences, Asan Medical Center, University of Ulsan College of Medicine, Seoul 05505, Korea; 3Department of Biomedical Sciences, Microbiology, University of Ulsan College of Medicine, Seoul 05505, Korea; sunkimnkt@daum.net (M.Y.K.); doloref@naver.com (W.S.C.); rainyis0603@gmail.com (E.Y.); gywjddl231@hanmail.net (H.J.L.); 4Stem Cell Immunomodulation Research Center (SCIRC), Asan Medical Center, University of Ulsan College of Medicine, Seoul 05505, Korea

**Keywords:** natural killer cell, GSK-3α, NKG2D, NKp30, BCR-ABL1

## Abstract

**Simple Summary:**

Glycogen synthase kinase-3 (GSK-3) is a serine/threonine protein kinase that has gained considerable interest as a therapeutic target for cancer due to its key involvement in growth arrest and apoptosis of tumor cells. Moreover, GSK-3, especially GSK-3β, limits the activation of NK cells, key innate effectors in cancer immunosurveillance, triggered by diverse activating receptors. However, the role of GSK-3 in the regulation of activating ligands on target cells that confer susceptibility to NK cells remains unclear and is the aim of this study. Here, we provide evidence that GSK-3α primarily restrains the expression of ligands for activating receptors such as NKG2D, NKp30 but not DNAM-1, thereby reducing target susceptibility to NK cells. Thus, our results suggest a distinct role of GSK-3 isoforms in target cells vs NK cells for regulating NK cell reactivity and GSK-3α inhibition as a relevant strategy to enhance target susceptibility to NK cells.

**Abstract:**

Natural killer (NK) cells are innate cytotoxic lymphocytes that provide early protection against cancer. NK cell cytotoxicity against cancer cells is triggered by multiple activating receptors that recognize specific ligands expressed on target cells. We previously demonstrated that glycogen synthase kinase (GSK)-3β, but not GSK-3α, is a negative regulator of NK cell functions via diverse activating receptors, including NKG2D and NKp30. However, the role of GSK-3 isoforms in the regulation of specific ligands on target cells is poorly understood, which remains a challenge limiting GSK-3 targeting for NK cell-based therapy. Here, we demonstrate that GSK-3α rather than GSK-3β is the primary isoform restraining the expression of NKG2D ligands, particularly ULBP2/5/6, on tumor cells, thereby regulating their susceptibility to NK cells. GSK-3α also regulated the expression of the NKp30 ligand B7-H6, but not the DNAM-1 ligands PVR or nectin-2. This regulation occurred independently of BCR-ABL1 mutation that confers tyrosine kinase inhibitor (TKI) resistance. Mechanistically, an increase in PI3K/Akt signaling in concert with c-Myc was required for ligand upregulation in response to GSK-3α inhibition. Importantly, GSK-3α inhibition improved cancer surveillance by human NK cells in vivo. Collectively, our results highlight the distinct role of GSK-3 isoforms in the regulation of NK cell reactivity against target cells and suggest that GSK-3α modulation could be used to enhance tumor cell susceptibility to NK cells in an NKG2D- and NKp30-dependent manner.

## 1. Introduction

Natural killer (NK) cells are cytotoxic innate lymphocytes and act as the first-line cellular barrier in cancer immunosurveillance [1,2,3]. NK cells rapidly kill a broad spectrum of tumor cells while sparing healthy cells through direct cytolysis and the production of cytokines (e.g., IFN-γ, TNF-α), without prior sensitization. Unlike major histocompatibility complex (MHC)-restricted killing of tumor cells by cytotoxic T cells, NK cells rely on an array of innate activating and inhibitory receptors for tumor cell recognition and elimination. The activating receptors that trigger NK cell effector functions include NKG2D, DNAM-1, NKp30, NKp44, and NKp46, each of which has a unique ligand specificity. As key antitumor effector cells that do not cause graft-versus-host disease, NK cells are considered a safe off-the-shelf cellular therapeutic for cancer immunotherapy [4,5,6]. Accumulating evidence has shown NK cell-based immunotherapy to be efficacious in some cancers, particularly hematologic malignancies, but it is often associated with a transient clinical benefit and eventual disease progression in many clinical settings. This limited success highlights the need to develop rational therapeutic strategies to achieve enhanced and durable clinical responses with NK cell therapy.

Glycogen synthase kinase-3 (GSK-3) is a serine/threonine protein kinase that is ubiquitously expressed and regulates diverse cellular functions including glycogen metabolism, cell death, and immune responses [7,8,9,10]. GSK-3 has two isoforms, GSK-3α (51 kDa) and GSK-3β (47 kDa), that are encoded by separate genes and differ significantly in the N- and C-terminal region [11]. Despite their redundancy in certain cellular functions [12], these isoforms also play non-overlapping roles, as demonstrated by distinct phenotypes in GSK-3α- and GSK-3β-deficient mice [13,14,15,16]. GSK-3β has the most substrates of any known kinase, with over 500 proposed substrates based on an in silico analysis [17] and approximately 100 confirmed substrates involved in key cellular pathways [18,19,20]. In this regard, GSK-3 has gained considerable attention as a promising therapeutic target in cancer therapy due to its central role in growth arrest and apoptosis of tumor cells and its specific targeting enabled by potent pharmacological inhibitors [19,21,22]. Accordingly, pharmacologic inhibition of GSK-3 leads to growth arrest or apoptosis in diverse leukemic cells [23,24,25]. Moreover, we previously demonstrated that GSK-3, especially GSK-3β, negatively regulates NK cell activity triggered by diverse activating receptors, including NKG2D and NKp30 [26]. In support, pharmacologic inhibition or genetic knockdown of GSK-3β enhances NK cell functions against K562 cells, a human chronic myeloid leukemia (CML) cell line. Similarly, pharmacologic or genetic inactivation of GSK-3β restores the cytotoxicity of NK cells against acute myeloid leukemia (AML) cells [27], suggesting the benefit of GSK-3 inhibition in leukemia treatment.

In addition, the expression of ligands for NK activating receptors is an important determinant of NK cell reactivity against cancer cells. To date, the role of GSK-3 in the regulation of activating ligands on leukemia cells remains unclear, given the results of a previous study that only used pharmacological agents for GSK-3 inhibition and only assessed the limited ligands (i.e., MICA/B and PVR/CD155) for NKG2D and DNAM-1 [28]. NKG2D recognizes MHC class I-like molecules, including ULBP1–6 as well as MICA/B, while the ligands for DNAM-1 are PVR/CD155 and nectin-2/CD112 [1,29,30]. In addition to these, NKp30 is a potent activating receptor that contributes to NK cell surveillance of leukemia cells and binds to B7-H6 on diverse tumor cells [31,32]. In this study, we sought to assess the role of GSK-3 isoforms in the NK cell susceptibility of CML cells that are sensitive or resistant to targeted therapy. Using both pharmacologic and genetic inactivation of GSK-3, we investigated the potential mechanisms through which GSK-3 isoforms regulate the expression of ligands for diverse NK cell activating receptors and whether such GSK-3 inhibition promotes the anti-leukemic activity of NK cells, including in a xenograft mouse model of leukemia clearance that relies on human NK cell activity.

## 2. Materials and Methods

### 2.1. Cells and Reagents

Human primary samples were obtained from healthy donors after informed consent in accordance with protocols approved by the Institutional Review Board (IRB) of Asan Medical Center and the Declaration of Helsinki. Peripheral blood mononuclear cells (PBMCs) from healthy donors were isolated using lymphocyte separation medium (MP Biomedicals, Santa Ana, CA, USA). Human NK cells were isolated from PBMCs by negative selection with an NK cell isolation kit (StemCell Technologies, Vancouver, BC, Canada). Isolated cell populations were 97–99% CD3^−^CD56^+^, as assessed by flow cytometry, and were expanded according to a protocol described below. PBMCs, purified NK cells, or expanded NK cells were used as effector cells for functional assays after 24 h of incubation with human recombinant IL-2 (rIL-2) (200 U/mL; Roche, Basel, Switzerland). The human CML blast crisis (BC) cell line KCL-22 was cultured in RPMI 1640 (Gibco/Thermo Fisher Scientific, Waltham, MA, USA) supplemented with 10% FBS (Gibco). The human tyrosine kinase inhibitor (TKI)-resistant CML-BC cell line KCL-22M, which harbors the BCR-ABL1 point mutation T315I, was cultured in RPMI 1640 supplemented with 10% FBS and 2 μM imatinib mesylate (IM) (Sigma-Aldrich, St. Louis, MO, USA). The presence of one *BCR-ABL1* allele containing the T315I mutation and one wild-type allele of the c-*ABL* gene in the KCL-22M cell line was confirmed using PCR amplification and direct sequencing of genomic DNA, as previously described [33]. Thiadiazolidinone (TDZD-8), lithium chloride (LiCl), LY294002, and 10058-F4 were from Calbiochem (San Diego, CA, USA). The FarRed cell proliferation kit and CSFE cell proliferation kit were from Invitrogen/Thermo Fisher Scientific.

### 2.2. NK Cell Expansion

Primary human NK cells isolated from PBMCs were expanded as previously described [34], with some modifications. PBMCs (1.5 × 10^6^ cells) were incubated in a 24-well tissue culture plate with 100 Gy-irradiated K562-mb15-41BBL cells (1 × 10^6^ cells) in Stem Cell Growth medium (SCGM; CellGenix, Freiburg, Germany) supplemented with 10% FBS and 10 U/mL rIL-2. The half volume of medium was exchanged every 2 days with fresh medium containing 10 U/mL rIL-2. After a week, residual T cells were depleted with a CD3 positive selection kit (StemCell Technologies. The remaining cells were incubated in SCGM supplemented with 10% FBS, 100 U/mL rIL-2, and 5 ng/mL rIL-15 (PeproTech, Rocky Hill, NJ, USA) for 2 additional weeks, with a half-medium exchange every 2 days. NK cell populations in PBMCs expanded on average 2000 fold by a total of 3 weeks. The expanded cell populations were 96–99% CD3^−^CD56^+^, as assessed by flow cytometry.

### 2.3. Antibodies

Antibodies (Abs) used for analysis of NK cell receptors and signaling molecules were detailed in Appendix A. Horseradish peroxidase (HRP)-conjugated anti-mouse and anti-rabbit secondary Abs were from Santa Cruz Biotechnology (Santa Cruz, CA, USA).

### 2.4. RT-PCR

To assess the effect of GSK-3 inhibition on ligand expression in KCL-22M cells, total RNA was isolated using the RNeasy kit (QIAGEN, Hilden, Germany) and cDNA was synthesized from 1 μg of RNA using the ReverTra Ace qPCR RT kit (Toyobo, Osaka, Japan), according to the manufacturer’s instructions. The PCR primers used were detailed in Appendix A.

### 2.5. RNA Interference

For siRNA-mediated knockdown of GSK-3α and/or GSK-3β, KCL-22 or KCL-22M cells were transfected with 300 pmol of specific siRNA using the Amaxa Nucleofector II system (Lonza, Basel, Switzerland). A total of 1.5 × 10^6^ cells were resuspended in 100 μL of Amaxa kit solution V, mixed with siRNA, and transfected using the program T-020. The cells were transfected for 48 h at 37 °C, washed, and assayed as indicated. The siRNAs specific for GSK-3α, GSK-3β, and c-Myc were obtained from IDT (Newark, NJ, USA) and had the following sequences: GSK-3α, 5′-CAA UAU UGU GAG GCU GAG AUA CUT T-3′ (sense) and 5′-AAA GUA UCU CAG CCU CAC AAU AUU GCA-3′ (antisense); GSK-3β, 5′-AAG AAU CGA GAG CUC CAG AUC AUG A-3′ (sense) and 5′-UCA UGA UCU GGA GCU CUC GAU UCU UAA-3′ (antisense); and c-Myc, 5′-CUA CAG CGA GUU AGA UAA AGC CCC GAA-3′ (sense) and 5′-CGG GGC UUU AUC UAA CUC GCU GUA G-3′ (antisense). A second set of siRNAs for c-Myc with the following sequences was used: 5′-CAA UUU GAG GCA GUU UAC AUU AUG GCU-3′ (sense) and 5′-CCA UAA UGU AAA CUG CCU CAA AUT G-3′ (antisense). Comparable results were obtained with both sets of c-Myc siRNA, and the results shown in the paper were obtained with the first set.

KCL-22M cells with stable knockdown of GSK-3α (KCL-22M shGSK-3α) were generated by transducing KCL-22M cells with the lentiviral GSK-3α shRNA construct. Lentiviral particles were produced by co-transfection of the 293TN cell line (Systems Biosciences, Palo Alta, CA, USA) with the pLKO.1-puro vector containing a validated shRNA sequence targeting GSK-3α (MISSION^®^ shRNA TRCN0000038682 or TRCN0000039766; Sigma-Aldrich) or a non-targeting control shRNA (SHC016; Sigma-Aldrich) and the pPACKH1 packaging plasmid mix (Systems Biosciences) using X-tremeGENE 9 (Roche). The medium was changed after 24 h, and virus-containing supernatant was collected after another 24 h. KCL-22M cells were transduced with lentivirus-containing supernatant in the presence of 10 μg/mL polybrene by spin-infection [34] and then selected with 3 μg/mL puromycin (2 days after transduction). After two rounds of puromycin selection, cells were analyzed for the knockdown of GSK-3α by immunoblotting and used for further experiments. Comparable results were obtained with both of the GSK-3α shRNAs, and the results shown in the paper were obtained with TRCN0000039766.

### 2.6. NK Cell Degranulation Assay

The cytotoxic degranulation of NK cells was determined by measuring the cell surface expression of CD107a, as previously described [35]. Briefly, IL-2-stimulated PBMCs or primary expanded NK cells (1 × 10^5^ cells) were mixed with an equal number of KCL-22 M cells in 96-well V-bottom culture plates (Corning Costar, NY, USA) and incubated for 2 h at 37 °C. For the blockade of NK activating receptors, Fc receptors on IL-2-stimulated PBMCs or primary expanded NK cells were blocked with human Fc Receptor Binding Inhibitor (eBioscience, San Diego, CA, USA) and then incubated with 20 μg/mL control IgG1 or Abs to the indicated NK activating receptors for 30 min at 4 °C prior to mixing with target cells. The cell pellets were resuspended in flow cytometry buffer (phosphate-buffered saline [PBS] with 1% FBS) and stained with anti-human CD3-PerCP, anti-human CD56-PE, and anti-human CD107a-FITC Abs for 35 min in the dark at 4 °C. Lymphocytes were gated on forward and side scatter characteristics, and the CD107a expression on CD3^−^CD56^+^ NK cells was analyzed by flow cytometry using a FACS Accuri C6 (BD Biosciences, Franklin Lakes, NJ, USA) and FlowJo software (ver.10, Treestar, Ashland, OR, USA).

### 2.7. NK Cell Intracellular IFN-γ Production Assay

IL-2-stimulated PBMCs (1 × 10^5^ cells) were stimulated with an equal number of KCL-22M cells for 1 h at 37 °C. Then, brefeldin A (GolgiPlug; BD Biosciences) and monensin (GolgiStop; BD Biosciences) were added, and followed by incubation for an additional 5 h, for a total of 6 h. The cells were first stained with anti-human CD3-PerCP and anti-human CD56-PE antibodies for 30 min in the dark at 4 °C. Samples were then washed twice with FACS buffer and incubated in BD Cytofix/Cytoperm solution (BD Biosciences) for 20 min in the dark at 4 °C. The cells were then washed twice with BD Perm/Wash buffer (BD Biosciences), stained with anti-human IFN-γ-FITC overnight in the dark at 4 °C, washed again, and analyzed by flow cytometry gated on CD3^−^CD56^+^ NK cells.

### 2.8. NK Cell Cytotoxicity Assay

For the europium-based cytotoxicity assay, KCL-22M cells were loaded with 40 μM BATDA reagent (Perkin Elmer, Waltham, MA, USA) for 30 min at 37 °C. Cells were then washed in medium containing 1 mM sulfinpyrazone (Sigma-Aldrich), resuspended at 1 × 10^6^ cells/mL in the medium, and incubated with primary expanded NK cells or purified NK cells in the presence of sulfinpyrazone for 2 h at 37 °C. Plates were mixed briefly and centrifuged at 1400 rpm for 5 min. Supernatant (20 μL) was incubated with 200 μL of 20% europium solution (Perkin Elmer) in 0.3 M acetic acid for 5 min, and target cell lysis was detected on a VICTOR X4 multi-label plate reader (Perkin Elmer).

### 2.9. Immunoblot Analysis

KCL-22M cells depleted of GSK-3 were washed with ice-cold PBS and lysed with a lysis buffer [50 mM Tris–HCl (pH 7.5), 150 mM NaCl, 1% Triton X-100, 5 mM EDTA, 1 mM NaVO_3_, 50 mM NaF, 1 mM PMSF, and protease inhibitor cocktail (Thermo Fisher Scientific)] for 30 min on ice. Cell lysates were then centrifuged to remove cell debris, including the nuclei, and supernatants were harvested. The protein concentration was determined with the Micro BCA protein assay kit (Pierce, Appleton, WI, USA). Lysates were diluted with 4× NuPAGE LDS sample buffer (Invitrogen) containing 50 mM DTT. Equal amounts of protein from each sample were resolved on 8% SDS-PAGE gels and transferred onto PVDF membranes (Millipore, Burlington, MA, USA) in transfer buffer [25 mM Tris, 192 mM glycine, 20% (*v*/*v*) methanol]. Membranes were blocked with 5% skim milk in TBS-T (TBS with 0.1% Tween 20) for 1 h at room temperature (RT), incubated with primary Abs overnight at 4 °C, and then incubated with HRP-conjugated secondary Abs for 1 h at RT. Blots were developed with SuperSignal West Pico (Pierce) and signals were detected using an LAS-4000 system (Fujifilm, Midtown West, Tokyo, Japan). All uncropped western blot figures can be found in Appendix A. 

### 2.10. In Vivo Leukemia Clearance Assay

To assess whether GSK-3α inhibition affects cancer surveillance by NK cells in vivo, we used a lymphoma clearance assay [36], modified to compare the killing of different tumors by human NK cells in the same immune-deficient mice. Briefly, 9–10-week-old NOD/ShiLtJ-*Rag2^em1AMC^Il2rg^em1AMC^* (NRG) mice were purchased from JA BIO (Suwonsi, Gyeonggido, Korea) and injected intraperitoneally (i.p.) with 200 U/mL rIL-2-activated primary expanded NK cells (1 × 10^6^ cells). KCL-22M shControl cells were labeled with a carboxyfluorescein succinimidyl ester (CFSE) (3 µM; Invitrogen), while KCL-22M shGSK-3α cells were labeled with FarRed (1 µM; Invitrogen). The cells were mixed in a 1:1 ratio (1 × 10^6^ cells per each cell type) and injected i.p. into NRG mice. After 4 h, peritoneal cells were collected and analyzed by flow cytometry, and rejection of KCL-22M shGSK-3α cells relative to KCL-22M shControl cells was calculated as follows: ratio of residual cancer cells = residual KCL-22M shGSK-3α cells (%)/residual KCL-22M shControl cells (%).

### 2.11. Statistical Analysis

Each experiment was performed in duplicate and repeated independently at least three times. Two groups were compared using two-tailed Student’s *t*-tests. Differences between multiple groups were analyzed by one-way analysis of variance (ANOVA). All data were analyzed using GraphPad Prism software (ver.5.00, GraphPad Software, Inc., San Diego, CA, USA). Statistical significance was defined as *p* < 0.05, and the degree of significance is indicated as follows: * *p* < 0.05, ** *p* < 0.01, and *** *p* < 0.001.

## 3. Results

### 3.1. Pharmacologic Inhibition of GSK-3 in TKI-Resistant CML Cells Enhances Their Expression of NKG2D Ligands and Susceptibility to NK Cells

Previously, we showed that GSK-3β acts as a negative regulator of multiple NK activation signals, mainly NKG2D [26]. Moreover, we provided evidence that NK cells could efficiently kill CML cells harboring a point mutation (T315I) that renders them resistant to all approved BCR-ABL1 TKIs, which remain a major clinical challenge [37]. We hypothesized that GSK-3 inhibitors might hold promise in the treatment of leukemia, including those with multi-TKI resistance, by promoting cancer immunosurveillance in addition to their direct anticancer activity. To address this possibility further, we investigated the effect of GSK-3 inhibition in TKI-resistant CML cells on the expression of ligands for NK cell activating receptors, with a focus on ligands for NKG2D. To this end, KCL-22M cells, a human CML-blast crisis cell line harboring the T315I mutation in the BCR-ABL1 kinase domain [33], was used. This cell line triggered NK cell cytotoxic degranulation in an NKG2D-dependent manner, as revealed by blockade of NKG2D with a specific Ab (Figure 1A).

GSK-3 was blocked with two different GSK-3 inhibitors, TDZD-8 (a GSK-3β-specific inhibitor) and LiCl (a pan-GSK-3 inhibitor). Thereafter, the surface expression of NKG2D ligands (NKG2DL), including MICA/B, ULBP1, ULBP2/5/6, and ULBP3, was assessed by flow cytometry. LiCl induced a clear upregulation of ULBP2/5/6, and, to a marginal extent, MICA/B on KCL-22M cells after 48 h of treatment, whereas TDZD-8 failed to upregulate ULBP2/5/6 (Figure 1B,C). Accordingly, LiCl but not TDZD-8 significantly increased the susceptibility of KCL-22M cells to NK cell cytotoxicity, as assessed by NK cell degranulation in IL-2-stimulated PBMCs (Figure 1D) and NK cell cytotoxicity in IL-2-stimulated purified NK cells (Appendix A). The upregulation of NKG2DL following LiCl treatment was confirmed at the level of gene expression with a moderate increase in MICA mRNA, corroborating the previous study [28], and a strong increase in ULBP2 mRNA (Appendix A). Collectively, although not striking, GSK-3 inhibition led to the moderate upregulation of NKG2DL, particularly ULBP2, and enhanced the susceptibility of TKI-resistant KCL-22M cells to NK cell cytotoxicity. Moreover, the selective effect of LiCl but not TDZD-8 in such a context raised the possibility of an isoform-specific role of GSK-3 in NKG2DL regulation.

### 3.2. GSK-3α Selectively Regulates NKG2DL Expression and Cytolysis by NK Cells

Next, the individual contributions of GSK-3 isoforms to the regulation of NKG2DL in TKI-resistant CML cells was investigated using siRNAs specific for GSK-3α or GSK-3β to rule out potential off-target effect of GSK-3 inhibitor and probe the specific role of GSK-3. A selective and significant knockdown of GSK-3α and/or GSK-3β was detected in the KCL-22M cells after transfection with the corresponding siRNAs (Figure 2A).

Knockdown of GSK-3α but not GSK-3β caused a notable upregulation in the levels of ULBP2/5/6 but no other ULBPs in KCL-22M cells (Figure 2B). In addition, the effect of GSK-3α knockdown on ULBP2/5/6 upregulation was nullified by combined knockdown of GSK-3β, suggesting a reciprocal effect of GSK-3 isoforms on the regulation of ULBP2/5/6. In comparison, the levels of MICA/B were moderately increased by individual knockdown of GSK-3α or GSK-3β, and further upregulated by their combined knockdown. As expected, quantitative real-time PCR confirmed the enhanced expression of ULBP2, and, to a lesser extent, ULBP1 and MICA by GSK-3α knockdown and upregulation of MICA by GSK-3β knockdown (Appendix A). Thus, these results suggest non-redundant roles for GSK-3α and GSK-3β in the regulation of NKG2DL, particularly ULBP2/5/6. Of interest, knockdown of GSK-3α had similar effects on the regulation of ULBP2/5/6 and MICA/B in KCL-22 cells harboring wild-type BCR-ABL1 (Appendix A), suggesting that GSK-3 regulates NKG2DL expression independently of oncogenic BCR-ABL1 mutations conferring TKI resistance.

We next assessed whether such a differential regulation of NKG2DL by GSK-3 isoforms on KCL-22M cells affects their susceptibility to NK cell-mediated lysis. To this end, primary NK cells from healthy donors were prepared using cytokines and a feeder-based expansion protocol adopted for use in clinical trials, and then used as effector cells. Knockdown of GSK-3α but not GSK-3β rendered KCL-22M cells more susceptible to NK cell-mediated lysis (Figure 2C), correlating with the increased levels of NKG2DL, particularly ULBP2/5/6. Consistently, NK cell degranulation against KCL-22M cells was increased more significantly by knockdown of GSK-3α than GSK-3β (Figure 2D). This increased degranulation was largely dependent on the NKG2D pathway, as determined by Ab-mediated blockade of NKG2D. Likewise but to a lesser extent, NK cell IFN-γ expression was significantly upregulated after depletion of GSK-3α but not GSK-3β in KCL-22M cells (Appendix A). These results suggest that expression of NKG2DL, especially ULBP2/5/6, is regulated by the GSK-3α isoform and preferentially controls the susceptibility of KCL-22M cells to NK cell-mediated cytotoxicity.

### 3.3. GSK-3α Regulates ULBP2/5/6 by PI3K/Akt- and c-Myc-Dependent Pathways

We next studied the mechanisms underlying the regulation of NKG2DL by GSK-3α, with a focus on ULBP2/5/6. PI3K and the mitogen-activated protein kinase (MAPK) ERK are among the signaling molecules regulated by GSK-3 [26,38,39] and are also important for NKG2DL upregulation [40,41,42]. Their role in the regulation of NKG2DL and consequent triggering of NK cell cytotoxicity was therefore investigated. GSK-3α knockdown enhanced the phosphorylation of Akt, which is downstream of PI3K activation, but did not affect the phosphorylation of ERK (Figure 3A).

Accordingly, the upregulation of ULBP2/5/6 and MICA/B following GSK-3α knockdown was nullified by treatment with LY294002 (LY), a pan-PI3K inhibitor (Figure 3B). Moreover, the same treatment reduced the baseline levels of ULBP2/5/6 and MICA/B on control KCL-22M cells, suggesting a broad impact of the PI3K pathway in NKG2DL regulation. Similar to the results with NKG2DL, we found a significant decrease in NK cell degranulation against GSK-3α-depleted KCL-22M cells upon LY treatment (Figure 3C).

We further investigated the involvement of c-Myc in NKG2DL regulation, given the close association of c-Myc with the PI3K/Akt pathway in Ras or oncogenic BCR-ABL1 signaling [43,44] and with the expression of ULBP1/2/3 in AML [45]. c-Myc is also a known substrate for GSK-3α, which phosphorylates c-Myc at a regulatory threonine residue and promotes its degradation via the ubiquitin/proteasome pathway [46,47]. As expected, GSK-3α knockdown led to the accumulation of c-Myc in KCL-22M cells (Figure 3D). The observed upregulation of NKG2DL after GSK-3α knockdown was markedly diminished by additional knockdown of c-Myc (Figure 3E), correlating with decreased NK cell degranulation against KCL-22M cells (Figure 3F). Similar results on NKG2DL regulation were obtained by treatment with a pharmacological inhibitor of c-Myc, 10058-F4 (F4) (Appendix A). These data collectively suggest that GSK-3α regulates NKG2DL expression in a manner dependent on PI3K/Akt and c-Myc activity.

### 3.4. GSK-3α Knockdown Enhances the Susceptibility of KCL-22M Cells to NK Cells In Vivo

We next addressed whether GSK-3α modulation also enhances the susceptibility of KCL-22M cells to NK cell-mediated killing in vivo. To this end, we first established KCL-22M cells with stable knockdown of GSK-3α after transduction with a lentiviral vector containing shRNA sequences targeting GSK-3α and antibiotic selection. The selected cells were referred to as KCL-22M-shGSK-3α or KCL-22M-shControl cells, respectively (Figure 4A).

Similar to the results with siRNA-mediated knockdown, KCL-22M-shGSK-3α cells exhibited enhanced upregulation of ULBP2/5/6 (Figure 4B) and increased susceptibility to NK cell cytolysis (Figure 4C), and triggered increased degranulation of NK cells (Figure 4D). For the in vivo study, we used a lymphoma clearance assay [36] modified to compare the clearance of different tumors by human NK cells in the same mice. Specifically, equal numbers of KCL-22M-shGSK-3α and KCL-22M-shControl cells were co-injected i.p. into immune-deficient NRG mice lacking NK cells after an injection of human primary NK cells (Figure 4E). To facilitate their identification, KCL-22M-shGSK-3α and KCL-22M-shControl cells were labeled with CellTrace FarRed and CFSE, respectively, in addition to forward scatter (FSC) vs. side scatter (SSC) gating (Appendix A). Analysis of the peritoneal fluid by flow cytometry revealed a significant increase in the clearance of KCL-22M-shGSK-3α cells over KCL-22M-shControl cells at 4 h post-injection (Figure 4F). The preferential clearance of KCL-22M-shGSK-3α cells was mediated by NK cells, as there was comparable recovery of the two cell populations in the absence of human primary NK cells. Although it requires further assessment of dead tumor cells among the peritoneal cells, these results suggest that GSK-3α inhibition in TKI-resistant KCL-22M cells enhances NK cell-mediated surveillance in vivo.

### 3.5. GSK-3α Knockdown Upregulates NKp30L (B7-H6) in Addition to NKG2DL

We finally investigated whether GSK-3α inhibition-mediated upregulation of activating ligands is confined to ligands for the NKG2D receptor or could be observed with ligands for other activating receptors, such as DNAM-1 and NKp30. The levels of DNAM-1L (CD155/PVR and CD112/Nectin-2) on KCL-22M-shGSK-3α cells were comparable to those on KCL-22M-shControl cells (Figure 5A), suggesting little or no effect of GSK-3α on DNAM-1L regulation. These results are consistent with a previous study showing no effect of GSK-3 inhibitors on CD155/PVR levels [28]. By contrast, we found a noticeable upregulation of NKp30L (B7-H6) on KCL-22M-shGSK-3α cells compared with KCL-22M-shControl cells, indicating an unexpected involvement of GSK-3α in the regulation of NKp30L. This notion was supported by the significant contribution of NKp30 to NK cell degranulation against KCL-22M-shGSK-3α cells upon receptor blockade (Figure 5B), in contrast to the marginal contribution of NKp30 to NK cell lysis of KCL-22M-shControl cells in the absence of GSK-3α inhibition (Appendix A) [37].

## 4. Discussion

Herein, we provide evidence that GSK-3, specifically GSK-3α, is a pivotal regulator of the expression of ligands for activating receptors, including NKG2D and NKp30, but not DNAM-1. Using a cell line model of TKI-resistant CML, we found that this regulation of NK cell activating ligands by GSK-3α was independent of an oncogenic BCR-ABL1 mutation conferring TKI resistance and occurred at the level of gene transcription as well as cell surface protein expression. Moreover, the finding that PI3K/Akt signaling and c-Myc activity were involved in this response provides new insights into the molecular mechanisms underlying the regulation of NK cell activating ligands and consequent target susceptibility by GSK-3α. Importantly, the therapeutic relevance of this finding was supported by our identification of preferential clearance of TKI-resistant KCL-22M cells depleted of GSK-3α by NK cells in vivo. Thus, our results suggest that GSK-3α is likely the primary GSK-3 isoform restraining the expression of NKG2DL, particularly ULBP2/5/6, and NKp30L by TKI-resistant KCL-22M cells and demonstrate that GSK-3α inhibition appears a relevant strategy to enhance target cell susceptibility to NK cell-mediated lysis. In support, similar regulation by GSK-3α but not GSK-3β of the expression of ULBP2/5/6 among other NKG2DL and NKp30L (B7-H6) and target susceptibility to NK cells was observed with K562 cells, a human CML cell line despite the difference in the pattern of ligand expression (Appendix A).

GSK-3 includes two isoforms, GSK-3α and GSK-3β, which control diverse cellular functions in both redundant and isoform-specific manners, depending on the context [7,48,49]. Previously, we demonstrated that GSK-3β but not GSK-3α negatively regulates the functions of human NK cells triggered by multiple activating receptors, including NKG2D, 2B4, and NKp30, and against K562 CML cells [26]. Thus, the dominant regulation of the activating ligands by GSK-3α, as shown here, highlights the differential roles of GSK-3α and GSK-3β in cancer surveillance by NK cells, further supporting the notion of isoform-specific functions of GSK-3. Inhibition of GSK-3α upregulated the expression of NKG2DL, particularly ULBP2/5/6 and to a lesser extent MICA/B and ULBP1, and NKp30L (B7-H6), but not DNAM-1L (CD155/PVR and CD112/Nectin-2) on TKI-resistant KCL-22M cells, leading to enhanced susceptibility to NK cell-mediated lysis. GSK-3β knockdown also induced a moderate upregulation of MICA/B but downregulated the levels of ULBP2/5/6, suggesting distinct regulation of activating ligands by different GSK-3 isoforms. Evaluation of MICA/B and ULBP2/5/6 at the level of gene expression revealed a selective regulation of ULBP2 by GSK-3α but a common involvement of both isoforms in the regulation of MICA but not MICB. This observation is consistent with a previous study showing upregulation of MICA but not MICB on multiple myeloma by pharmacological inhibitors of GSK-3 targeting GSK-3β or both isoforms [28].

A mechanistic study indicated that NKG2DL upregulation by GSK-3α inhibition could be attributed to an increase in PI3K/Akt signaling and c-Myc activity. Consistent with this, knockdown of GSK-3α increased the levels of Akt phosphorylation and c-Myc, both of which were required for the upregulation of activating ligands. PI3K/Akt is considered a key signaling pathway in the regulation of NKG2DL such as ULBP1/2/3 and MICA/B [40,41]. Recent studies also support this notion by showing the upregulation of MICA/B on multiple tumor cell types via the PI3K/Akt pathway [50,51]. Intriguingly, PI3K/Akt signaling has been implicated in phosphorylation of GSK-3α and GSK-3β at regulatory serine residues, resulting in GSK-3 inhibition [20,52]. Given our result that GSK-3α inhibition increased Akt phosphorylation, we speculate that there is a feedback loop between the PI3K/Akt pathway and GSK-3α for the regulation of NK activating ligands on leukemia cells. We also reported that knockdown of GSK-3β in NK cells resulted in increased phosphorylation of Akt, potentiating NK cell functions [26]. Thus, such a cross-regulation between PI3K/Akt and GSK-3 isoforms appears to be a common mechanism underlying the regulation of NK cell reactivity against leukemia cells according to the context (i.e., activating ligand expression vs. NK cell function). Moreover, c-Myc is often dysregulated and overexpressed in many different tumor types, including leukemia [53,54], and it drives the expression of NKG2DL in a mouse leukemia model [55] and the expression of ULBP1/2/3 in human AML [45]. GSK-3α is known to phosphorylate c-Myc at threonine 58, promoting its degradation via the ubiquitin/proteasome pathway [46,47]. This regulation is consistent with our data showing increased levels of c-Myc upon GSK-3α depletion in KCL-22M cells and a previous study showing c-Myc accumulation by pharmacological blockade of pan-GSK-3 in *KRAS* mutant tumors [56]. However, the involvement of other regulatory mechanism(s) cannot be excluded and merits further investigation, given that GSK-3 is a central hub of many signaling pathways.

Importantly, NKG2DL were not the only ligands regulated by GSK-3α. We found that the NKp30L B7-H6 is also upregulated by GSK-3α depletion, contributing to the enhanced sensitivity of KCL-22M cells to NK cells. B7-H6 is frequently expressed on the surface of different tumor cells and has recently been shown to be regulated by a mechanism involving c-Myc induction [57]. Given the increased levels of c-Myc upon GSK-3α knockdown, we speculate that c-Myc is potentially linked to B7-H6 regulation by GSK-3α, which merits further investigation. Moreover, as NKG2DL are frequently downregulated or absent on leukemia blasts and leukemic stem cells [3,58,59], the upregulation of B7-H6 by GSK-3α inhibition could provide an additional therapeutic target to enhance NK cell reactivity against leukemia cells. Moreover, the involvement of other ligands regulated by GSK-3α cannot be excluded, which warrants further study.

An array of GSK-3 inhibitors with different isoform specificities are available and have been shown to induce apoptosis or growth arrest of various leukemia cell lines and primary leukemia cells [60,61,62,63]. A primary concern in the potential clinical application of GSK-3 inhibitors, particularly pan-GSK-3 inhibitors, is their possible stabilization and activation of β-catenin, which can promote oncogenic transformation [12,21,63,64]. This safety concern has encouraged continued interest in developing a strategy to target GSK-3 in an isoform-specific manner. This notion is supported by recent studies showing a potential clinical benefit of selective genetic targeting of GSK-3α over GSK-3β, which inhibited AML progression [60] and myeloid transformation [64] without affecting β-catenin. In support of this, BRD0705, a GSK-3α-selective inhibitor, induced myeloid differentiation and impaired colony formation in AML cells without affecting normal hematopoietic cells and β-catenin stability [63]. In this regard, targeted modulation of GSK-3α may hold promise as a viable strategy to improve NK cell reactivity against leukemia cells in addition to its direct anti-leukemic effects, given the upregulation of NKG2DL and NKp30L in leukemia cells upon GSK-3α inhibition. Moreover, an increase in activating ligands also occurred in TKI-resistant CML cells carrying the T315I mutation of BCR-ABL1, one of the major cause of mortality in CML patients [65,66]. Thus, targeted inhibition of GSK-3α may provide an alternative therapeutic strategy against refractory/relapsed CML cells with TKI-resistance and possibly other leukemia cells in the context of NK cell-based therapy.

## 5. Conclusions

Altogether, we found that GSK-3α appears the primary isoform restraining the expression of a subset of ligands for activating receptors such as NKG2D, NKp30 but not DNAM-1, thereby regulating the susceptibility of TKI-resistant CML cells to NK cells in vitro and in vivo. Moreover, the finding of PI3K/Akt signaling and c-Myc in such a context provides an insight into the molecular mechanism of the ligand upregulation by GSK-3α inhibition. Although further validation is required using different cell types and primary leukemic blasts, the present study may suggest the distinct role of GSK-3 isoforms in the regulation of ligands for NK activating receptors and GSK-3α modulation as a potential strategy for enhancing anti-tumor reactivity of NK cells.

## Figures and Tables

**Figure 1 cancers-13-01802-f001:**
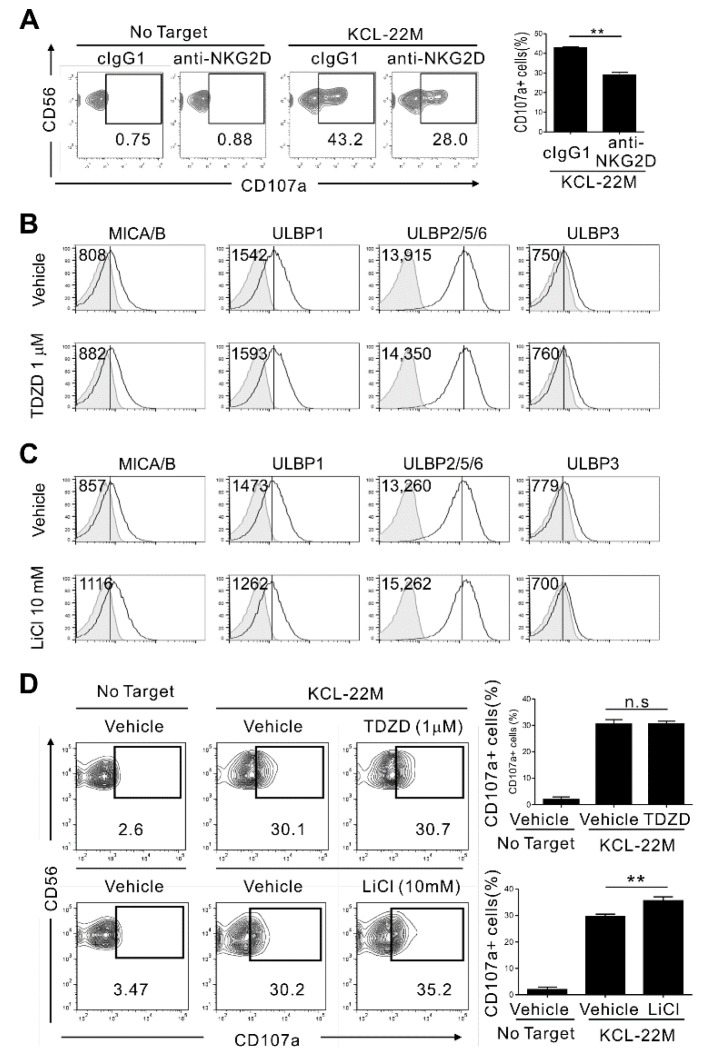
Inhibition of GSK-3 enhances NKG2DL expression and NK cell-mediated cytotoxicity against KCL-22M cells. (**A**) KCL-22M cells were pretreated with IM (5 μM) for 48 h, washed, and then mixed with expanded primary NK cells preincubated with a monoclonal Ab to the NKG2D receptor (20 μg/mL) for 2 h. Degranulation of NK cells was measured by the surface expression of CD107a on CD3^−^CD56^+^ NK cells. Representative flow cytometry profile (left) and summary graph (right) demonstrating the percentage of CD107a^+^ NK cells. (**B**,**C**) The surface expression of NKG2DL (MICA/B, ULBP1, ULBP2/5/6, and ULBP3) was analyzed by flow cytometry on KCL-22M cells treated with the GSK-3 inhibitor TDZD-8 (1 μM; GSK-3β-selective) (**B**) or LiCl (10 mM; pan-GSK-3) (**C**) for 48 h. (**D**) KCL-22M cells were pretreated with GSK-3 inhibitors as in (**B**,**C**) for 48 h, and then mixed with IL-2-stimulated PBMCs for measurement of degranulation. Representative flow cytometry profile (left) and summary graph (right) demonstrating the percentage of CD107a^+^ on CD3^−^CD56^+^ NK cells. Values represent the means ± s.d. Data are representative of at least three independent experiments. ** *p* < 0.01.

**Figure 2 cancers-13-01802-f002:**
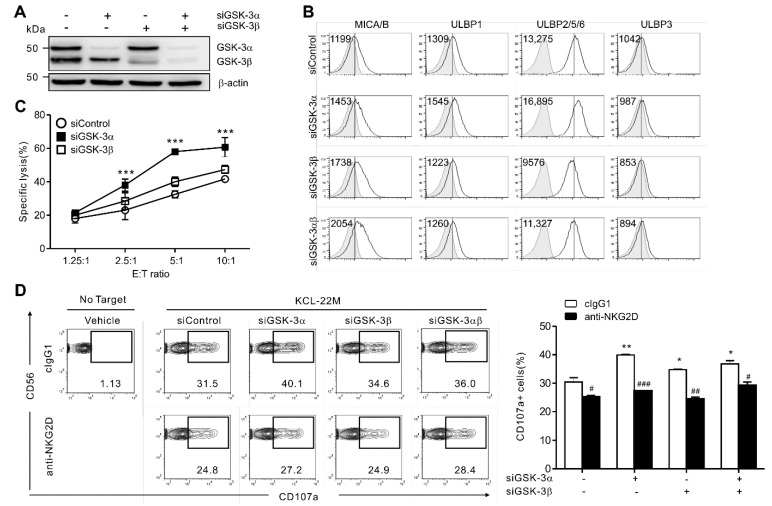
GSK-3α selectively regulates NKG2DL expression on TKI-resistant KCL-22M cells and their cytolysis by NK cells. (**A**) Total cell lysates of KCL-22M cells transfected with control siRNA or siRNAs specific for GSK-3α and/or GSK-3β for 48 h were immunoblotted for GSK-3α/β and actin. (**B**) After siRNA-mediated knockdown, KCL-22M cells were analyzed for the surface expression of NKG2DL (MICA/B, ULBP1, ULBP2/5/6, and ULBP3) by flow cytometry. (**C**) NK cell cytotoxicity against KCL-22M cells depleted of GSK-3α and/or GSK-3β was assessed using a europium-based cytotoxicity assay with expanded primary NK cells at the indicated effector to target (E:T) cell ratios. (**D**) Degranulation of NK cells against KCL-22M cells depleted of GSK-3α and/or GSK-3β was measured using expanded primary NK cells in the absence or presence of NKG2D blockade. Representative flow cytometry profile (left) and summary graph (right) demonstrating the percentage of CD107a^+^ on CD3^−^CD56^+^ NK cells. Values represent the means ± s.d. Data are representative of at least three independent experiments. *^,#^
*p* < 0.05; **^,##^
*p* < 0.01; ***^,###^
*p* < 0.001.

**Figure 3 cancers-13-01802-f003:**
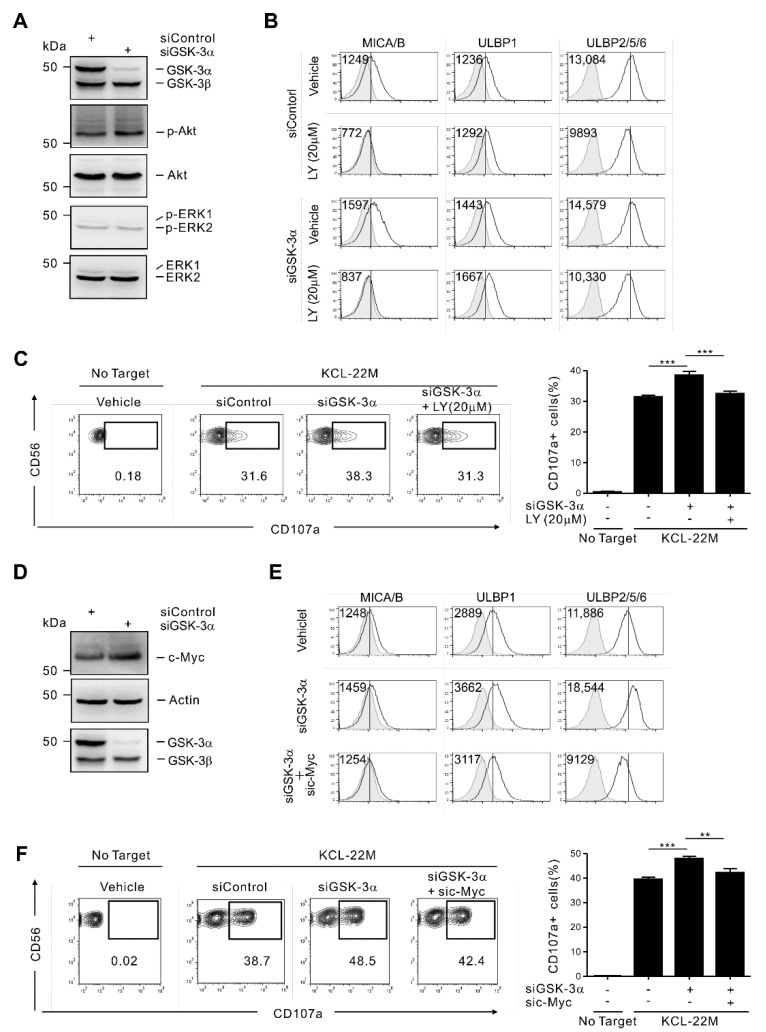
GSK-3α knockdown enhances NKG2DL expression via Akt signaling and c-Myc activity. (**A**) Total cell lysates from KCL-22M cells transfected with control siRNA or GSK-3α-specific siRNAs for 48 h were immunoblotted for GSK-3α/β, p-Akt, Akt, p-ERK1/2, and ERK1/2. (**B**) Analysis of NKG2DL surface expression on KCL-22M cells after siRNA-mediated knockdown of GSK-3α in the absence or presence of the PI3K inhibitor LY (20 μM) for 48 h. (**C**) Degranulation of NK cells in PBMCs against KCL-22M cells after siRNA-mediated knockdown of GSK-3α in the absence or presence of the PI3K inhibitor LY (20 μM) for 48 h. Representative flow cytometry profile (left) and summary graph (right) demonstrating the percentage of CD107a^+^ on CD3^−^CD56^+^ NK cells. (**D**) Total lysates of KCL-22M cells transfected with control siRNA or a GSK-3α-specific siRNA for 48 h were immunoblotted for c-Myc, β-actin, and GSK-3α/β. (**E**) Analysis of NKG2DL surface expression on KCL-22M cells after siRNA-mediated knockdown of GSK-3α or GSK-3α and c-Myc. (**F**) Degranulation of NK cells in PBMCs against KCL-22M cells depleted of GSK-3α or GSK-3α and c-Myc. Representative flow cytometry profile (left) and summary graph (right) demonstrating the percentage of CD107a^+^ on CD3^−^CD56^+^ NK cells. Values represent the means ± s.d. Data are representative of at least three independent experiments. ** *p* < 0.01; *** *p* < 0.001.

**Figure 4 cancers-13-01802-f004:**
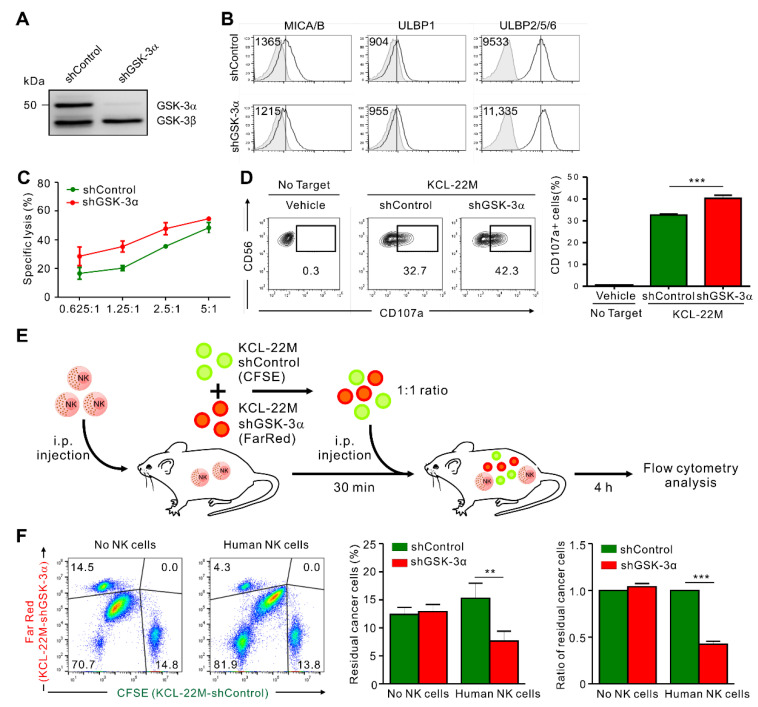
GSK-3α knockdown enhances NK cell-mediated protection against T315I BCR-ABL KCL-22M cells. (**A**) Total cell lysates from KCL-22M cells transduced with lentivirus expressing control shRNA or an shRNA specific for GSK-3α were immunoblotted for GSK-3α/β. (**B**) Analysis of NKG2DL surface expression on KCL-22M cells transduced with control shRNA (shControl) or with stable knockdown of GSK-3α (shGSK-3α). (**C**) NK cell cytotoxicity against KCL-22M cells with control shRNA (shControl) or stable knockdown of GSK-3α (shGSK-3α) was assessed using a europium-based cytotoxicity assay at the indicated effector to target (E:T) cell ratios. (**D**) Degranulation assay with expanded primary NK cells against KCL-22M cells transduced with control shRNA (shControl) or with stable knockdown of GSK-3α (shGSK-3α). Representative flow cytometry profile (left) and summary graph (right) demonstrating the percentage of CD107a^+^ on CD3^−^CD56^+^ NK cells. (**E**) In vivo leukemia rejection protocol. Immunodeficient NRG mice lacking B, T, and NK cells received an intraperitoneal injection with vehicle or expanded primary NK cells. For target cell preparation, KCL-22M-shControl cells were labeled with CFSE, while KCL-22M-shGSK-3α cells were labeled with FarRed. A 1:1 target cell mix was then injected intraperitoneally into NRG mice 30 min post-injection of NK cells, and the rejection of KCL-22M-shGSK-3α cells relative to KCL-22M-shControl cells in the peritoneal cavity was measured by flow cytometry after 4 h. (**F**) Flow cytometry of CFSE-stained KCL-22M-shControl and FarRed-stained KCL-22M-shGSK-3α cells injected at a ratio of 1:1 (Input) into the mice (*n* = 6 per group), followed by analysis of CFSE and FarRed in cells recovered from the peritoneal cavity of recipient mice 4 h later (Output) to assess NK cell killing activity in vivo. Representative flow cytometry profile (left), summary graph showing percentage (middle), and ratio (right) of residual cancer cells. Values represent the means ± s.d. Data are representative of at least three independent experiments. ** *p* < 0.01; *** *p* < 0.001.

**Figure 5 cancers-13-01802-f005:**
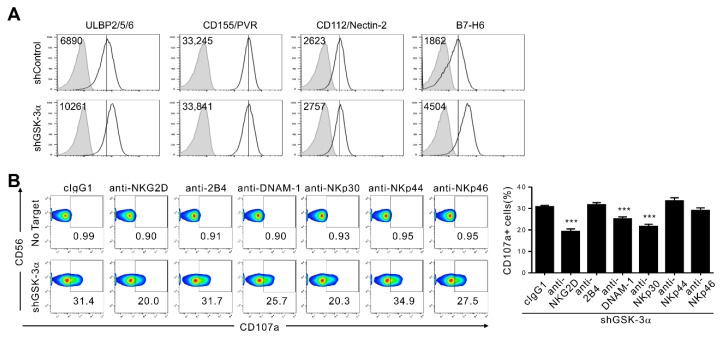
GSK-3α knockdown induces the upregulation of NKp30L (B7-H6) but not DNAM-1L (CD155 and CD112). (**A**) Analysis of ULBP2/5/6, ligands of the NKG2D receptor; CD155/PVR and CD112/nectin-2, ligands of the DNAM-1 receptor; and B7-H6, the ligand of the NKp30 receptor, on KCL-22M cells transduced with control shRNA (shControl) or with stable knockdown of GSK-3α (shGSK-3α). (**B**) KCL-22M cells transduced with lentivirus expressing an shRNA specific for GSK-3α were mixed with expanded primary NK cells preincubated with blocking Abs to the indicated receptors (20 μg/mL) for 2 h. Degranulation was measured by the expression of CD107a on CD3^−^CD56^+^ NK cells. Representative flow cytometry profile (left) and summary graph (right) demonstrating the percentage of CD107a^+^ NK cells. Values represent the means ± s.d. Data are representative of at least three independent experiments. *** *p* < 0.001.

## Data Availability

The data presented in this study are available on reasonable request from the corresponding author.

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
