# Peer review of "GSK-3α Inhibition in Drug-Resistant CML Cells Promotes Susceptibility to NK Cell-Mediated Lysis in an NKG2D- and NKp30-Dependent Manner"

_cancers, 2021, doi:10.3390/cancers13081802_

Round 1
Reviewer 1 Report
The manuscript is well written and clear. I found that the materials and methods are well explained in details. The quality of the presentation of results is high and data are clearly explained in graphs and histograms. The discussion is supported by results shown in literature, and the authors highlight the possible clinical application of GSK-3 inhibitors in AML and CML. The paper needs a minor spell check.
Author Response
Re) We thank the reviewer for appreciation of our work. In the revision, we carefully checked the spell and corrected if any error including the label in Figure 5A (shControl).
Reviewer 2 Report
In this manuscript, Kim et al investigate the effects of GSK-3alpha inhibition in drug-resistant CML cells to NK cell-mediated lysis. Although the authors present a technically impressive dataset, there are several criticisms and comments that might help to improve the paper.
1) The authors use either total PBMC or highly pure expanded NK cells. Why were the ex vivo NK cells not purified? This would have clarified the contribution of the NK cells to the readout without leaving doubts about the potential role of other cell types (for example NK-like T cells).
2) A direct comparison of cytotoxic and degranulation activities of PBMC and expanded NK cells is missing. Here again, it would have been useful to compare directly ex vivo purified NK cells with K562-IL15-4.1BBL-expanded NK cells. It would also be interesting to indicate the cell yield of the IL2-activated PBMC and the expanded pure NK cells.
3) Line 109: "Germany" and not "German".
4) Lines 116-148: these two paragraphs are extremely difficult to read. It would be much better to list the antibodies and the primers in two different tables with, for the antibodies, the following columns: antibody to which antigen, antibody clone, antibody isotype, antibody purified or fluorochrome-conjugated, supplier.
5) In the experiments with the NK cell expansion, the authors describe that they changed the medium every two days. How was this medium change performed? By harvesting and centrifuging the cells?
6) Figure 1B: no effects; Figure 1C: minor effect on ULBPs; LiCl: minor effect; overall, the conclusions to figure 1 are hyperbole (lines 288-290).
7) Line 407: one cannot apply the terms "increased susceptibility" to degranulation
8) Line 425: the authors claim "little or no effect of GSK-3alpha on DNAM1L regulation". However, in Figure 5B, they show a significant effect on this ligand when applying GSK-3alpha inhibition. This discrepancy should be explained.
9) Lines 455-457: before making such a claim, additional target cell lines, namely other CML lines and cell lines of different histological types known to express NKG2DL and/or NKp30L, should be tested. Furthermore, it would strengthen the paper if a positive control for cytotoxicity, and particularly K562, which is also a CML cell line, would be used in all experiments, and if primary leukemic blasts (inhibitor-sensitive and inhibitor-resistant ones ) would be added too. Although it is true that inhibitor-resistant cell lines are a clinical problem, the potential of NK cell-based therapeutic approaches goes far beyond this.
Author Response
In this manuscript, Kim et al investigate the effects of GSK-3alpha inhibition in drug-resistant CML cells to NK cell-mediated lysis. Although the authors present a technically impressive dataset, there are several criticisms and comments that might help to improve the paper.
1) The authors use either total PBMC or highly pure expanded NK cells. Why were the ex vivo NK cells not purified? This would have clarified the contribution of the NK cells to the readout without leaving doubts about the potential role of other cell types (for example NK-like T cells).
Re) We first used total PBMCs to assess the effect of target cell GSK-3 inhibition on NK cell functions because of the limitation of the supply of highly pure primary NK cells and the possible contribution of other cell types in such context. Given the positive results, we next used highly pure expanded NK cells to determine the direct contribution of NK cells to the readouts (Figure 2C for cytotoxicity and Figure 2D for degranulation). We preferred to use highly pure expanded NK cells in our study, given their preparation according to the protocol adopted in clinical trials and the focus of this study on the relevance of target cell GSK-3 inhibition in NK cell-based therapy. However, according to the reviewer’s comment, we conducted an additional experiment using ex vivo purified NK cells stimulated with IL-2 to assess the effect of GSK-3 inhibition (LiCl and TDZD-8) on cytotoxicity. Consistently, treatment of KCL-22M cells with LiCl but not TDZD-8 significantly enhanced cytotoxicity of ex vivo purified NK cells (New supplementary figure S1). In this regard, the text has been revised accordingly to indicate the increased susceptibility of KCL-22M cells to NK cells upon GSK-3 inhibition [Lines 91~92; Lines 187~188; Lines 265~266].
2) A direct comparison of cytotoxic and degranulation activities of PBMC and expanded NK cells is missing. Here again, it would have been useful to compare directly ex vivo purified NK cells with K562-IL15-4.1BBL-expanded NK cells. It would also be interesting to indicate the cell yield of the IL2-activated PBMC and the expanded pure NK cells.
Re) In our study and experience, we observed that cytotoxic activities of K562-IL15-4.1BBL-expanded NK cells are about 2~3 fold higher than those of ex vivo purified NK cells (Figure 2C vs. New supplementary figure S1) and are 10~20 fold higher than those of PBMCs (data not shown) at the same effector to target cell ratios. In addition, we observed that NK cell populations in PBMCs did not expand significantly after 24 h of IL-2 stimulation but expanded on average 2,000 fold by day 21 according to the K562-IL15-4.1BBL feeder-based expansion protocol. In this regard, the text has been revised accordingly to indicate the cell yield of NK cells from PBMCs after expansion [Line 115].
3) Line 109: "Germany" and not "German".
Re) We thank the reviewer for commenting this mistake, which has been corrected in the revised method for NK cell expansion (change of “German” to “Germany”) [Line 109].
4) Lines 116-148: these two paragraphs are extremely difficult to read. It would be much better to list the antibodies and the primers in two different tables with, for the antibodies, the following columns: antibody to which antigen, antibody clone, antibody isotype, antibody purified or fluorochrome-conjugated, supplier.
Re) As suggested by the reviewer, the lists of antibodies and the primers used in our study are presented in separate table formats including the required information for antibodies (New supplementary Table 1 for antibodies and New supplementary Table 2 for primers) [Lines 118~119 for antibodies; Lines 126~127 for primers].
5) In the experiments with the NK cell expansion, the authors describe that they changed the medium every two days. How was this medium change performed? By harvesting and centrifuging the cells?
Re) NK cell expansion culture was continued by carefully replacing half-volume of the medium with fresh medium every two days. As requested by the reviewer, the medium change is detailed in the revised method for NK cell expansion [Line 110; Line 114].
6) Figure 1B: no effects; Figure 1C: minor effect on ULBPs; LiCl: minor effect; overall, the conclusions to figure 1 are hyperbole (lines 288-290).
Re) As commented by the reviewer, the effect of GSK-3 inhibitor on NKG2D ligand expression and NK cell cytotoxicity was not striking. However, we observed a consistent and moderate increase in the expression of NKG2D ligands, particularly ULBP2/5/6, (Figure 1C) and significant upregulation of NK cell cytotoxicity (Figure 1D and New supplementary figure S1) upon target cell GSK-3 inhibition (LiCl but not TDZD-8). To rule out potential off-target effect of GSK-3 inhibitors and probe the specific role of GSK-3, we therefore used siRNAs specific for GSK-3a and GSK-3b for GSK-3 inhibition. In this regard, we have toned down the effect of GSK-3 inhibitor in such a context and commented on the necessity to use isoform-specific siRNAs for GSK-3 inhibition [Lines 261~262; Lines 269~270; Lines 277~278].
7) Line 407: one cannot apply the terms "increased susceptibility" to degranulation
Re) We thank the reviewer for commenting this error, which has been corrected in the revised manuscript (change to “triggered increased degranulation of NK cells”) [Line 390].
8) Line 425: the authors claim "little or no effect of GSK-3alpha on DNAM1L regulation". However, in Figure 5B, they show a significant effect on this ligand when applying GSK-3alpha inhibition. This discrepancy should be explained.
Re) KCL-22M cells triggered NK cell cytotoxic degranulation in an NKG2D and DNAM-1-dependent manner even in the absence of GSK-3a inhibition, which was described as reference citation [Kim N et al., Cancers, 12:1923, 2020, PMID: 32708713] and data not shown in the previous manuscript. Upon GSK-3a inhibition, we observed a clear increase in the contribution of NKG2D and NKp30, rather than DNAM-1, to NK cell degranulation (Figure 5B), consistent with the upregulation of NKG2DL and NKp30L but not DNAM-1L in KCL-22M-shGSK-3a cells compared with KCL-22M-shControl cells (Figure 5A). We are sorry for the confusion not showing the data with KCL-22M-shControl cells and therefore have included the data assessing NK cell degranulation against KCL-22M-shControl cells upon the blockade of NK cell receptors (New supplementary figure S8) in the revised manuscript [Lines 416~418].
9) Lines 455-457: before making such a claim, additional target cell lines, namely other CML lines and cell lines of different histological types known to express NKG2DL and/or NKp30L, should be tested. Furthermore, it would strengthen the paper if a positive control for cytotoxicity, and particularly K562, which is also a CML cell line, would be used in all experiments, and if primary leukemic blasts (inhibitor-sensitive and inhibitor-resistant ones) would be added too. Although it is true that inhibitor-resistant cell lines are a clinical problem, the potential of NK cell-based therapeutic approaches goes far beyond this.
Re) As suggested by the reviewer, we also assessed the effect of GSK-3 inhibition on the expression of NKG2DL and NKp30L on K562 cells, a human CML cell line, and their susceptibility to NK cells. Similar to the results with KCL-22M cells, knockdown of GSK-3a rather than GSK-3b caused an upregulation of ULBP2/5/6 among other NKG2DL and B7-H6 on K562 cells and a significant increase in NK cell degranulation against K562 cells (New supplementary figure S9) [Lines 442~446]. Unfortunately, besides limited time given for revision, patients with TKI-sensitive and TKI-resistant CML blasts are not accessible under our IRB approval that limits the study to the healthy donors. Accordingly, the requirement for further study with different cell types and primary leukemic blasts has been discussed in the revised manuscript [Lines 527~528]. Moreover, as suggested by the reviewer, we have toned down the therapeutic importance of GSK-3a in NK cell-based cancer therapy [Line 439; Lines 441~442; Lines 518~519; Lines 528~530].
Reviewer 3 Report
In the manuscript “GSK-3α Inhibition in Drug-Resistant CML cells Promotes Sus-2 ceptibility to NK Cell-Mediated Lysis in an NKG2D- and 3 NKp30-Dependent Manner” important questions are raised, including the role of GSK3 isoforms in regulating ligands to NK cell receptors and how these affect NK cell cytolytic activity against certain tumors. Nevertheless, data included in this manuscript showing moderate changes and experimental design lacks the ability to answer all questions raised.
- Figure 1A,B there is a clear NKG2D effect however not complete blocking, any other synergistic receptors like NCRs as you show they play role in GSK3i?
- In Figure 1B and 1C histograms for vehicle staining for ULBP2/5/6 is different, which shouldn’t be unless these were done on different passaged cells or different days then it wouldn’t be a fair control. For MICA/B and ULBP3 there was absolutely no difference unless wrong graphs are added.
- The CD107a show very little change and not always correlate with specific lysis. Specific killing assays should be performed to confirm treatments effect.
- The conclusion of first result paragraph thus is not supported by the moderate changes shown in the first figure. To support the conclusion, a stronger inhibitor or titration of the inhibitors’ concentration might be needed
- Figure 2C is the difference between siCtrl and the isoforms significant?
- Figure2D right and in general no information about n biological/experimental replicates, which make it difficult to evaluate data.
- Apparently, the KD has very little effect on cytokine production but higher on the NK cell cytotoxicity, thus the focus should be on the cytolytic pathway. Are there any changes in granzyme B or perforin?
- How were dead tumor cells in vivo determined, a gating strategy for eliminating NK cells ( and other cells) and showing dead cell marker should be included.
- In Figure 5B blockade of the different pathways should be done to the siCtrl in comparison to shGSK3a. Current data don’t allow for full evaluation.
Author Response
In the manuscript “GSK-3α Inhibition in Drug-Resistant CML cells Promotes Susceptibility to NK Cell-Mediated Lysis in an NKG2D- and 3 NKp30-Dependent Manner” important questions are raised, including the role of GSK3 isoforms in regulating ligands to NK cell receptors and how these affect NK cell cytolytic activity against certain tumors. Nevertheless, data included in this manuscript showing moderate changes and experimental design lacks the ability to answer all questions raised.
Figure 1A,B there is a clear NKG2D effect however not complete blocking, any other synergistic receptors like NCRs as you show they play role in GSK3i?
Re) KCL-22M cells triggered NK cell degranulation in a manner largely dependent on NKG2D and DNAM-1 but not NCRs, which was described as reference citation [Kim N et al., Cancers, 12:1923, 2020, PMID: 32708713] and data not shown in the previous manuscript. Upon GSK-3a inhibition, we observed a significant contribution of NKp30, besides NKG2D and, to a lesser extent, DNAM-1, to NK cell degranulation (Figure 5B), correlating with the regulation of their cognate ligands (Figure 5A). To highlight the involvement of other receptors besides NKG2D in such context, we have included the data assessing NK cell degranulation against KCL-22M-shControl cells (no GSK-3a inhibition) upon the blockade of NK receptors (New supplementary figure S8) in the revised manuscript, together with the data obtained from KCL-22M-shGSK-3a cells (GSK-3a inhibition) (Figure 5B). However, the involvement of other ligands for NK activating receptors in conferring the susceptibility to NK cells and their regulation by GSK-3a cannot be excluded. Accordingly, this point, together with the requirement for further study, has been discussed in the revised manuscript [Lines 416~418; Lines 499~500].
In Figure 1B and 1C histograms for vehicle staining for ULBP2/5/6 is different, which shouldn’t be unless these were done on different passaged cells or different days then it wouldn’t be a fair control. For MICA/B and ULBP3 there was absolutely no difference unless wrong graphs are added.
Re) In the previous manuscript, we presented the data in Figure 1B (TDZD-8 treatment) and Figure 1C (LiCl treatment) as a representative figure although they were performed on different days, as commented by the reviewer. We also performed the experiment with LiCl and TDZD-8 in the same day and obtained the similar results. As requested by the reviewer, the original data in Figure 1B and 1C histograms have been replaced with the new data done in the same day (Revised Figure 1B and 1C).
For MICA/B and ULBP3, the effect of LiCl on their upregulation was marginal (MICA/B) and almost not shown (ULBP3) compared with ULBP2/5/6 (Revised Figure 1C). Although not striking, we consistently observed marginal upregulation of MICA/B upon LiCl treatment, which was consistent with the increase in MICA mRNA (Supplementary figure S2) and corroborated the previous study [Fionda C et al., J Immunol, 190:6662, 2013, PMID: 23686482]. In this regard, the text has been revised accordingly to indicate the marginal effect of LiCl on MICA/B but no effect on ULBP3 [Lines 261~262].
The CD107a show very little change and not always correlate with specific lysis. Specific killing assays should be performed to confirm treatments effect.
Re) According to the reviewer’s comment, we conducted an additional experiment using purified NK cells stimulated with IL-2 to assess the effect of GSK-3 inhibition (LiCl and TDZD-8) on NK cell-mediated cytolysis. Similar to the results with PBMCs, treatment of KCL-22M cells with LiCl but not TDZD-8 significantly enhanced cytotoxicity of purified NK cells (New supplementary figure S1). In this regard, the text has been revised accordingly to indicate the increased susceptibility of KCL-22M cells to NK cell-mediated killing upon LiCl-mediated GSK-3 inhibition [Lines 265~266].
The conclusion of first result paragraph thus is not supported by the moderate changes shown in the first figure. To support the conclusion, a stronger inhibitor or titration of the inhibitors’ concentration might be needed
Re) As commented by the reviewer, the effect of GSK-3 inhibitors on NKG2D ligand expression and NK cell cytotoxicity was not striking. Although we tested various concentrations of the GSK-3 inhibitors, the presented data were optimal with the indicated dose of LiCl (10 mM) and TDZD-8 (1 mM) in our study. As we cannot rule out potential off-target effect of GSK-3 inhibitors, we therefore used siRNAs specific for GSK-3a and GSK-3b to probe the specific role of GSK-3. In this regard, we have toned down the effect of GSK-3 inhibitor in such a context and commented on the necessity to use isoform-specific siRNAs for GSK-3 inhibition [Lines 261~262; Lines 269~270; Lines 277~278].
Figure 2C is the difference between siCtrl and the isoforms significant?
Re) As requested by the reviewer, we have included statistical analysis in Figure 2C, demonstrating a significant increase in NK cell-mediated lysis upon knockdown of GSK-3a but not GSK-3b (Revised Figure 2C). In this regard, we have incorporated revised figure with statistical significance of the data.
Figure2D right and in general no information about n biological/experimental replicates, which make it difficult to evaluate data.
Re) We obtained all the results with at least three independent experiments and presented the data including Figure 2D as a representative one. As requested by the reviewer, we added the sentence pointing the experimental replicates of the data in the figure legends [Lines 256~257; Lines 289~290; Lines 359~360; Line 387; Line 427].
Apparently, the KD has very little effect on cytokine production but higher on the NK cell cytotoxicity, thus the focus should be on the cytolytic pathway. Are there any changes in granzyme B or perforin?
Re) We agree with the reviewer’s opinion that KCL-22M cells depleted of GSK-3a triggered a significant increase in NK cell cytotoxicity and, to a lesser extent, NK cell IFN-g expression. This preference for NK cell cytotoxicity corroborated a recent study of the requirement for less stimulation to induce cytotoxic degranulation than IFN-g expression [Fauriat C et al., Blood, 115:2167, 2010, PMID: 19965656]. We observed no significant change in granzyme B and perforin of NK cells in response to KCL-22M cells depleted of GSK-3a (data not shown). Thus, we anticipated that the increased susceptibility of KCL-22M cells after GSK-3a knockdown to NK cell cytolysis is primarily attributed to the upregulation of activating ligands such as NKG2DL and NKp30L. In this regard, the text has been revised accordingly to indicate the preferential control of NK cell cytotoxicity over IFN-g expression upon GSK-3a knockdown [Line 314; Line 317].
How were dead tumor cells in vivo determined, a gating strategy for eliminating NK cells (and other cells) and showing dead cell marker should be included.
Re) For the in vivo study, we adopted a widely used lymphoma clearance assay [Gascoyne DM et al. Nat Immunol, 10:1118, 2009, PMID: 19749763; Hyun YM et al., Sci Adv, 6:eabc4882, 2020, PMID: 33158867], modified to directly compare the killing of KCL-22M-shControl cells (no GSK-3a inhibition) and KCL-22M-shGSK-3a cells (GSK-3a inhibition) by human NK cells in the same immune-deficient NRG mice lacking T, B as well as NK cells. To facilitate their identification in harvested peritoneal cells by flow cytometry, KCL-22M-shControl cells and KCL-22M-shGSK-3α were labeled with CFSE (bright FL1) and CellTrace FarRed (bright FL4), respectively. In addition, we used forward versus side scatter (FSC vs SSC) gating closely matched to KCL-22M cells for the selective gating of injected KCL-22M cells rather than leukocytes including NK cells. Although we previously tried to identify dead tumor cells in vivo using LIVE/DEAD dye staining in harvested peritoneal cells by flow cytometry, it was unsuccessful in our hands, at least possibly due to the rapid clearance of dead cells by phagocytes. Thus, the assessment of tumor cell death in vivo by human NK cells warrants further investigation. This point, together with the requirement for further study, has been discussed in the revised manuscript [Lines 402~403]. Moreover, we have incorporated the gating strategy for identifying the CFSE-stained KCL-22M-shControl and FarRed-stained KCL-22M-shGSK-3a cells within the gate closely matched to KCL-22M cells (New supplementary figure S7) [Line 396~397].
In Figure 5B blockade of the different pathways should be done to the siCtrl in comparison to shGSK3a. Current data don’t allow for full evaluation.
Re) As requested by the reviewer, we have included the data assessing NK cell degranulation against KCL-22M-shControl cells (no GSK-3a inhibition) upon the blockade of NK receptors (New supplementary figure S8) in the revised manuscript, together with the data obtained from KCL-22M-shGSK-3a cells (GSK-3a inhibition) (Figure 5B). Comparison of these data revealed a clear increase in the contribution of NKG2D and NKp30, rather than DNAM-1, to NK cell degranulation upon GSK-3a knockdown (Figure 5B), consistent with the upregulation of NKG2DL and NKp30L but not DNAM-1L in KCL-22M-shGSK-3a cells compared with KCL-22M-shControl cells (Figure 5A) [Lines 416~418].
Round 2
Reviewer 3 Report
The work has been significantly improved from the last version and the new data added to this version has clarified and addressed most of my questions. There is a minor error in the text for fig 3F (row 340) that should be decreased instead of increase degranulation.